# Before & After: The Effect of EU's 2022 Code of Practice on Disinformation

## ABSTRACT

Over the past few years, the European Commission has made significant steps to reduce disinformation in cyberspace. One of those steps has been the introduction of the 2022 "Strengthened Code of Practice on Disinformation". Signed by leading online platforms, this Strengthened Code of Practice on Disinformation is an attempt to combat disinformation on the Web. The Code of Practice includes a variety of measures including the demonetization of disinformation, urging, for example, advertisers "to avoid the placement of advertising next to Disinformation content".

In this work, we set out to explore what was the impact of the Code of Practice and especially to explore to what extent ad networks continue to advertise on dis-/mis-information sites. We perform a historical analysis and find that, although at a hasty glance things may seem to be improving, there is really *no significant reduction in the amount of advertising relationships among popular misinformation websites and major ad networks*. In fact, we show that ad networks have withdrawn mostly from *unpopular* misinformation websites with very few visitors, but still form relationships with highly unreliable websites that account for the majority of misinformation traffic. To make matters worse, we show that ad networks continue to place advertisements of legitimate companies next to misinformation content. In fact we show that major ad networks place ads in almost 400 misinformation websites of our dataset.

## CCS CONCEPTS

• **Information systems** → *World Wide Web*; **Online advertising**.

**ACM Reference Format:**

Anonymous Author(s). 2024. Before & After: The Effect of EU's 2022 Code of Practice on Disinformation. In *Proceedings of ACM Conference (Conference'17)*. ACM, New York, NY, USA, 11 pages. https://doi.org/10.1145/nnnnnnn.nnnnnnn

## 1 INTRODUCTION

From the US presidential elections in 2016 [1] to the Brexit referendum [2], Covid-19 [3], the war in Ukraine [4] and the recent conflict in the Middle East [5], disinformation today is shaping major global events. It tears apart the fabric that holds our societies together by destroying people's faith in traditional news sources and undermining people's trust in governments, public institutions

and democratic processes [6]. Disinformation is usually designed to appeal to our worst impulses, fears and prejudices – all in an attempt to shape opinions or divide and conquer a society on the information battlefield. According to a recent survey, over 80% of EU citizens see fake news as an issue both for their country, and for democracy in general [7].

In an attempt to mitigate this threat, governments, tech firms and stakeholders have explored various methods to identify and curtail the spread of fake news. In 2022, the United Nations in their report [8] set out the relevant international legal framework, and discussed measures that States and technology enterprises reported to have taken to counter disinformation. In 2022, Google launched the Google's Digital News Innovation Fund that supported 662 digital news projects in Europe (total amount: 150M euros) [9]. A few years earlier, the European Union issued the 2018 Code of Practice on Disinformation, where representatives of online platforms, leading tech companies and players in the advertising industry (e.g., Facebook, Twitter, Google, Mozilla with Microsoft and TikTok following) agreed on a self-regulatory Code of Practice to address the spread of online disinformation [10]. The Code of Practice is based on 21 commitments in different domains, including the transparency in political advertising and, more importantly, demonetization of purveyors of disinformation.

A year after its implementation, the Code was already acknowledged globally as a pioneering framework [11]. An assessment of the Code was published in 2020 [12], showing that (i) the Code did provide a valuable framework for a structured dialogue between online platforms, thus, ensuring transparency and accountability of their policies on disinformation, but also (ii) a set of important gaps and shortcomings. In June 2022, the Commission issued the Strengthened Code of Practice on Disinformation (CoP) [13].

In this paper, we set out to explore the impact of the Strengthened CoP with regards to one of its most important commitments: the demonetization of disinformation spreading websites. We perform a historical analysis on misinformation websites in order to investigate what was the impact of the policy on their ad revenue by comparing their direct business relationships with ad networks.

The contributions of this work are summarized as follows:

(1) We discover that there is no significant reduction in the amount of advertising accounts with popular ad networks, after the Code of Practice on Disinformation came into effect.

(2) We demonstrate that after the CoP came into effect, popular ad networks have withdrawn mostly from unpopular disinformation websites, with hardly any visitors. On the contrary, we find that these ad networks still have connections with popular misinformation websites that account for the majority of misinformation traffic.

(3) Two years after the CoP was signed, signatories still accept 1 out of 3 misinformation websites as authorized ad inventory sellers.

(4) We show that ad networks do not substantially differentiate between unreliable and reliable news websites in their business relations and that they form business relationships with both kinds of websites.

(5) We establish that ad networks still serve ads to almost 400 misinformation websites (from our dataset) and facilitate the monetization of misinformation content.

(6) We find that ads of 23 Fortune 500 companies are served next to misinformation content, endangering their brand reputation and consumer trust.

(7) We make our list of approximately 2,500 misinformation news websites publicly available [14].

## 2 BACKGROUND

In the modern Web, when users visit a website, an automated programmatic process matches them with advertisers. Advertisers entrust Demand-Side Platforms (DSPs) to find the most suitable users to promote their brand or product, while website publishers authorize Supply-Side Platforms (SSPs) to manage their ad inventory and monetize their content. To eradicate various forms of ad fraud, stakeholders have introduced new standards.

`ads.txt:` Due to the complexity of the advertising ecosystem, bad actors were able to sell ad inventory of websites they did not control. To eliminate this, the Internet Advertising Bureau (IAB) Technology Laboratory introduced the `ads.txt` specification [15] that allows publishers to explicitly disclose who is authorized to sell the ad inventory of their website. This information is enclosed within a text file, publicly available at the root of the domain (snippet in Figure 1). Each `ads.txt` file contains comma-separated records with each record authorizing an entity to sell the ad inventory of the website. Each record contains: (i) the domain name of the advertising system (e.g., SSP), (ii) an identifier for the seller's account, (iii) the type of the account, and optionally (iv) a certification id for the advertising system. The type of the account represents the relationship between the account holder and the advertising system. A `DIRECT` account suggests that the website publisher also controls the advertising account and has a direct business contract with the ad system. For a `RESELLER` account, the website publisher has authorized a third-party entity to resell their ad inventory.

`sellers.json:` To increase transparency in the advertising ecosystem, the IAB Tech Lab also introduced the `sellers.json` standard [16]. This standard helps discover the identities of the entities that sell the ad inventory of websites and can help identify all intermediaries involved in ad inventory selling. The `sellers.json` standard is implemented as a JSON file published by advertising systems in order to disclose the inventory sellers they have approved within their ad system. Each approved seller is represented by a unique identifier inside this file. This is the same identifier that appears in the `ads.txt` file and maps to a single entity that is paid for the ad inventory it sells (example in Figure 1). Additionally, each identifier is described by a seller type. A `PUBLISHER` seller suggests that this account directly owns and controls the website whose ad inventory is sold through the advertising system and that the advertising system directly pays this entity. An `INTERMEDIARY` sellers indicates that is a reseller of ad inventory. Finally, a seller can be labeled as `BOTH`, acting as both a publisher and intermediary.

## 3 METHODOLOGY

### 3.1 Sources of Misinformation

To assemble a list of misinformation websites and study their behavior over time, we make use of publicly available datasets produced by academic publications, as well as by credible and acclaimed organizations. Our first sources is MediaBias/FactCheck (MBFC) [17], an organization that investigates bias and misinformation in online sources of information. By using human evaluators and a concrete and consistent methodology [18], they evaluate the factual accuracy of various media sources. We visit the MBFC website in October 2023 and extract labels and information about the bias, factual reporting, credibility, traffic volume and organization type for each evaluated website. We retain only websites that have a "LOW CREDIBILITY" score **and** a factual reporting of "MIXED", "LOW" or "VERY LOW" that form a list of 1,677 misinformation websites. We consciously exclude websites from the "Satire" category from this analysis, since they deliberately use humor and exaggeration in their published articles and do not attempt to deceive visitors. Such websites do not fall under the misinformation umbrella.

Next, we use the publicly available lists of previous academic works to extend our dataset of fake news websites as assembled in [19] consisting of 1,044 misinformation websites. Additionally, we extract 669 more misinformation sites from [20] that classifies news websites using independent sources (e.g., Politifact, Snopes). Finally, we make use of the *FakeNewsNet* tool used in [21] and download all news articles from the collected dataset. We exclude all data from the GossipCop fact-checking service because it has been discontinued and is no longer available. We extract 8 misinformation websites by filtering websites that have published at least 4 fake news articles (proven by Politifact [22]). We also filter out Facebook because we argue that it is neither a news publisher or news aggregator. Misinformation articles or posts shared in such a big platform should not classify the entire social network as a misinformation website.

We merge all the above lists and end up with a dataset of 2,469 distinct misinformation websites. In an attempt to foster academic research and give back to the academic community, we make our list of misinformation websites publicly available [14].

### 3.2 Fetching `ads.txt` and `sellers.json` files

Our study revolves around the analysis of `ads.txt` and `sellers.json`, two important files in the digital advertising that help combat ad fraud and increase transparency in the advertising ecosystem. Previous work has demonstrated that these files can reveal business relationships among websites and ad networks [19, 20, 23]. We first study the business relationships among ad networks and misinformation websites and how these have changed over the years and especially after the Code of Practice on Disinformation was signed. To that extent, we utilize `ads.txt` served by misinformation websites, and the `sellers.json` files served by popular ad networks. We crawl the list of approximately 2,500 misinformation websites on January 2024 and download 226,659 `ads.txt` entries coming from 1,132 misinformation websites. We make use of an open-source crawler [24] to collect and parse the respective `ads.txt` files. We find there are `ads.txt`

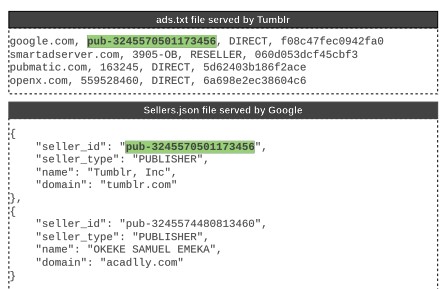
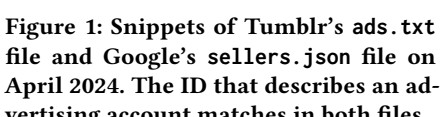

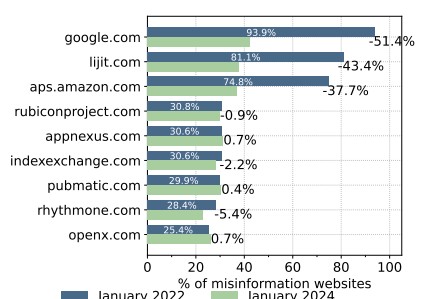
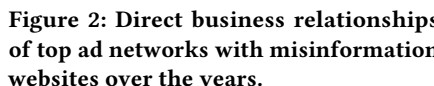

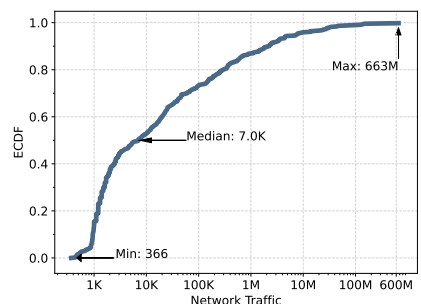

**Figure 1: Snippets of Tumblr's `ads.txt` file and Google's `sellers.json` file on April 2024. The ID that describes an advertising account matches in both files.**

**Figure 2: Direct business relationships of top ad networks with misinformation websites over the years.**

**Figure 3: Distribution of number of visits of selected misinformation websites on January 2024 (Note: $x$-axis is in log scale).**

records representing relationships with over 1,100 distinct ad networks, suggesting that ad networks still work with objectionable websites. To get a better understanding of the actual business relationships, we direct our analysis to `ads.txt` entries marked as `DIRECT`. The `ads.txt` specification defines that `DIRECT` accounts indicate the website owner (i.e., publisher) directly controlling the advertising account, and that there is a direct business contract between the publisher and the ad system [15]. Note that a website may provide bogus information in its `ads.txt` file. For example, it may claim that a legitimate advertiser sells ads on this website. On the other hand, an ad seller may also provide a fake `sellers.json` file claiming that it can put ads in high-profile websites to attract more customers. Although each individual file of the above two files may provide fake information, their intersection provides the truth. Indeed, if website A (in its `ads.txt` file) claims that it displays content from ad network B, **and** ad network B (in its `sellers.json` file) claims that it displays ads in website A, then this is true. In the rest of this work we take the intersection of these files. That is, we form a relation between A and B only if we find this relation both in `ads.txt` file of A and in `sellers.json` file of B.

## 3.3 Ad Collection

In order to study ads that are actually served on misinformation websites, as well as the ad networks that serve them, we develop a fully automated system that visits websites, emulates the behavior of a real user and clicks on ads.

First, we establish different user profiles (i.e., personas) to emulate real users and present a compelling profile to ad networks. We form different personas emulating different real-world users so that they receive ads from different advertisers, thus revealing a better picture of the ad ecosystem around misinformation. Additionally, we emulate users from various geographic locations in order to (potentially) get ads from both acclaimed companies and brands known worldwide, as well as local advertisers. We select Netherlands, Greece and USA as the different vantage points. We use a VPN service to set the location of our virtual users and before we begin forming the personas, we use two different geolocation services to ensure that these virtual users indeed appear to be in the specified country. In order to build a persona, we visit a set of specific

websites and expect that third parties will infer the user's gender and preferences based on the content they consume. We describe the methodology we follow to build user profiles in Appendix A.

After we have built each persona, we use this user profile to visit the list of misinformation websites. We utilize the open-source crawler presented in [25] and extend it to cover the needs of this work. The crawler visits both the landing page and a randomly selected internal page of each website and detects the advertisers in these. Prior work has demonstrated that internal pages can have significant differences in both structure and content (including ads and trackers) [26]. We extend the crawler to also detect online ads using cosmetic filters, based on Easylist [1] and uBlock Origin [2], two of the most popular ad-blocking projects. Additionally, to make the crawler appear as close to an actual user as possible, we use a headful browser with custom preferences (e.g. dark color scheme, window size, etc.). Finally, we make use of Consent-O-Matic [27], a reliable extension, to automatically accept all cookies in consent banners and ensure that ads are not blocked because the user has not interacted with the consent banner. We configure the extension to automatically accept all data processing purposes.

## 4 REGULATION IMPACT

We set out to explore if the Strengthened Code of Practice on Disinformation has had a meaningful impact on the monetization of misinformation content. We expect that the ecosystem will be improving because multiple signatories have committed to cutting the advertising revenue of disinformation spreaders [13]. Inspired by prior work [19], we study how direct business relationships of misinformation websites with ad networks have evolved over time.

We utilize the dataset of `ads.txt` and `sellers.json` files described in Section 3.2, as well as the one collected in [19] for news websites in December 2021. Arguably, publishers of fake news content could easily lie in their `ads.txt` files [20, 24]. To discover the actual business relationships, we focus on information coming from both the publishers and the ad systems. That is, we cross-reference the claims that publishers make in their `ads.txt` files with those of trustworthy ad networks in `sellers.json` files. We argue that

---

[1] https://easylist.to/
[2] https://ublockorigin.com/

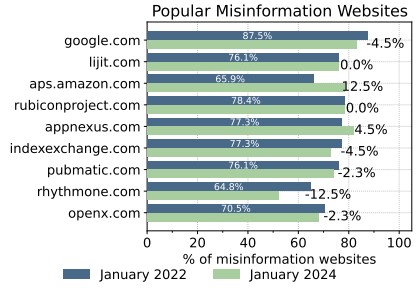

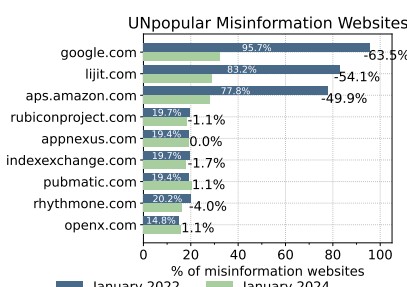

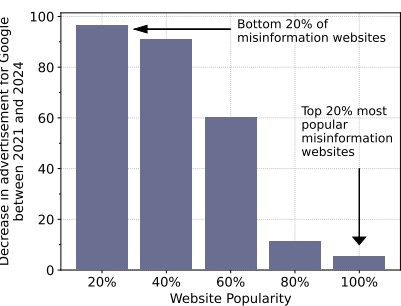

**Figure 4: Direct business relationships of top ad networks with popular misinformation websites over the span of 2 years.**

**Figure 5: Direct business relationships of top ad networks with unpopular misinformation websites after 2 years.**

**Figure 6: Difference in misinformation websites working with Google based on website network traffic.**

since these are conflicting sides of the same coin, the intersection of their claims is the ground truth. For each distinct DIRECT ads.txt entry, we review whether the respective publisher ID is also disclosed by the respective ad network. For instance, if *examplefakenews.com* has the entry adnetwork.com, 12345, DIRECT in its ads.txt file, we confirm that *adnetwork.com* also discloses the identifier 12345 in its sellers.json file. Again, we focus only on ads.txt records labeled as DIRECT because they indicate a direct business contract between the misinformation publisher and the advertising system. First, we study the behavior of the top ad networks and how their business relationships with misinformation websites have evolved over a period of 2 years. We focus on the intersection of these datasets and find that there exist 461 misinformation websites with an ads.txt file in both datasets and with at least one business relationship also disclosed by the respective ad network. We plot our findings in Figure 2.

At first glance, we observe that the majority of ad networks still (i.e., at January 2024) work with misinformation websites and that they have not pulled away from them. In fact, 6 out of the 9 most popular ad networks (i.e., rubiconproject.com to openx.com in Figure 2) have only blocked a very small percentage of misinformation websites (i.e., up to 5.4%). However, one can observe that there exist ad systems that evidently have stopped working with a substantial amount of misinformation websites. Indeed, Google, Lijit and Amazon seem to have dropped 51%, 43% and 37% of misinformation websites that they worked with 2 years ago. Although this drop is significant and seems to be in line with the CoP, a closer look reveals that these websites are insignificant and represent only a tiny portion of the misinformation traffic out there. To further investigate this point, we use the SimilarWeb analytics platform [28] to extract traffic data for misinformation websites. We successfully extract the network traffic of 95% of the misinformation websites for which we have collected their ads.txt files, both before and after the CoP was signed. In Figure 3, we plot the distribution of their network traffic in terms of number of visits on desktop and mobile during January 2024. Please note that the *x*-axis is in logarithmic scale, suggesting that there exist a lot of misinformation websites with low network traffic and a few highly popular that are able to attract millions of monthly visitors. We split these websites on the 80th percentile based on the observation that the top 20% are

popular websites, while the rest are unpopular.[3] The 80th percentile corresponds to a network traffic of approximately 290K monthly visits. Consequently, we find 88 misinformation websites which we label as "popular", and 351 labeled as "unpopular".

We perform a similar analysis to the one described in Figure 2 and plot the evolution of business relationships of the top ad networks with popular and unpopular misinformation websites in Figures 4 and 5, respectively. These figures paint a clear picture of the ad ecosystem. That is, even though it seems that some ad networks indeed no longer work with misinformation websites, they do so only for the unpopular ones. In fact, some of these websites have hardly any visitors and their effect on the dissemination of misinformation is limited, deeming them potentially insignificant. When it comes to the big players of misinformation, that are extremely popular and attract up to 100s of millions of visits on a monthly basis, we observe in Figure 4 that ad networks still work with them, even a year and half after the CoP was signed. Few top ad networks no longer have business relationships with a small percentage of the most popular misinformation websites, while we find ad networks such as Amazon and Microsoft (i.e., AppNexus) that now work with more popular misinformation websites compared to 2 years ago.

To further support our finding, we explore how business relationships with misinformation websites have been affected based on their popularity. As a case study, we focus on Google, the most popular ad network [24]. We group misinformation websites into equal-sized buckets based on their network traffic, expressed in monthly visits during January 2024. As in previous figures, to increase our confidence on the validity of these business relationships, we cross-reference these accounts with those that Google reports in their sellers.json and only report those accounts that match. That is, we only study advertising accounts that Google agrees are valid within their ad system. We then compute the difference in the misinformation websites that work with these ad networks between December 2021 and January 2024. We express this difference as a percentage increase $D = \frac{A-B}{|B|} \cdot 100$, where $A$ (i.e., After) is the number of misinformation websites having a direct business relationship with the ad network on 2024/01, while $B$ (i.e., Before) is

---

[3]The split 80-20 could have been made as 85-15, or 75-25. The point here is not to define an optimal threshold but to create two categories: the "popular" misinformation websites and the "unpopular" ones.

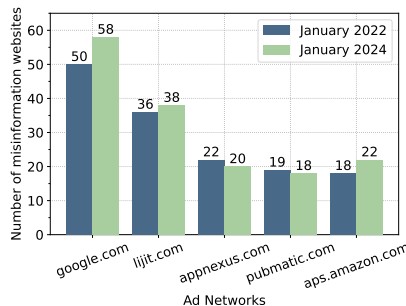

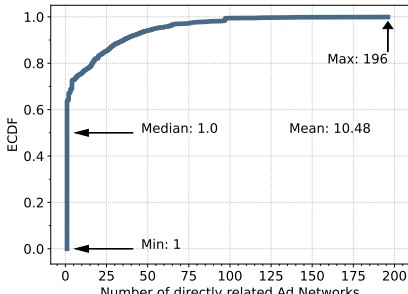

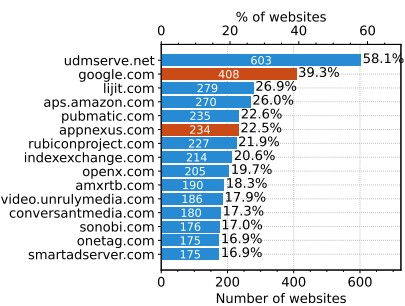

**Figure 7: Evolution of popular misinformation websites explicitly stated in `sellers.json` files of popular ad networks.**

**Figure 8: Distribution of the number of direct business relationships between a misinformation website and ad networks.**

**Figure 9: Direct business contracts with misinformation websites on July 2024 based on `ads.txt` and `sellers.json`. Red bars represent CoP signatories [29].**

| Ad Network | Websites | Spearman | p-value | Minimal |
|---|---|---|---|---|
| *google.com* | 502 | 0.64 | 5.021135e-57 | -72.62% |
| *lijit.com* | 429 | 0.63 | 1.333227e-47 | -73.57% |
| *aps.amazon.com* | 415 | 0.64 | 8.267313e-48 | -75.28% |

**Table 1: Correlation between misinformation websites traffic and progress of direct ad networks business contracts. The last column shows the percentage of low-traffic websites that no longer have business relationships after 2 years.**

for 2022/01. We plot our findings in Figure 6. The major difference in the number of misinformation websites that no longer have a verified account with Google is observed for less popular websites (i.e., left-most bar of the plot). For example, approximately 96% of the misinformation websites that are ranked on the bottom 20% based on their network traffic no longer have a direct business relationship with Google. On the other hand, we observe almost no difference for popular misinformation websites (i.e., right-most bar of the plot). Thus, we observe a discrepancy on how popular misinformation websites are treated, compared to unpopular ones.

Inspired by the finding of Figure 6, we study if there is a correlation between the network traffic of a misinformation website and whether an ad network stopped working with this website. For this, we make use of the traffic labels that MBFC assigns to news websites. MBFC estimates traffic data through the SimilarWeb platform, but also takes into consideration the number of subscribers for print media and market size for TV or Radio entities [18]. The labels assigned to each website are "Minimal", "Low", "Medium" or "High". We focus on the ad networks of Figure 2 that showed the greatest decrease in the number of misinformation websites they work with. We find that there is indeed a correlation between the network traffic of a misinformation website and if it declares a business contract with an ad network after two years. We present our findings in Table 1. The Spearman correlation coefficient is ~0.6 with a *p*-value close to zero for all of the three ad networks we study. In fact, we observe that over 70% of misinformation websites with minimal network traffic no longer have a business relationship with these ad networks after two years.

Recent work has demonstrated that misinformation websites often use dark pooling in order to pool their ad inventory with

that of lawful and acceptable websites [20, 24, 30]. In an attempt to prove explicit relationships between popular misinformation websites and popular ad networks, we study their `sellers.json` files. In Figure 7, we plot the number of misinformation websites from our list of 2,500 misinformation websites that are explicitly stated inside `sellers.json` files of popular ad systems. This is a definitive proof that ad networks still fund misinformation websites because they explicitly name them as verified ad inventory sellers.

We observe that the number of misinformation websites explicitly disclosed in the `sellers.json` files of the most prestigious and popular ad networks is somewhat stable. It is evident that the behavior of the top ad networks has not changed significantly over the last two years, even after the Code of Practice on Disinformation was signed. In fact, studying the standard deviation of the number of popular misinformation websites in `sellers.json` files (less than 2.5) shows that almost all networks have not stopped doing business with the misinformation websites. We provide examples of such relationships in Appendix B.

> **Finding 1:** Major ad networks still accept business relationships with misinformation websites which account for the majority of the misinformation traffic. After the CoP, these ad networks have ceased their collaboration mainly with the unpopular misinformation websites.

## 5 FINANCIAL INCENTIVES

The analysis of Section 4 suggests that misinformation websites still rely on popular ad networks to monetize their content. We cross reference the accounts declared in `ads.txt` files with those found in `sellers.json` ad networks, just like in Section 4 and find that there exist 985 (87%) misinformation websites with at least one substantiated business relationship with an ad network. That is, for 985 misinformation websites there exists an ad network that confirms the publisher account exists within its network and that it indeed belongs to a publisher. This demonstrates that misinformation websites are still able to get approved by ad networks and monetize their content through ad revenue.

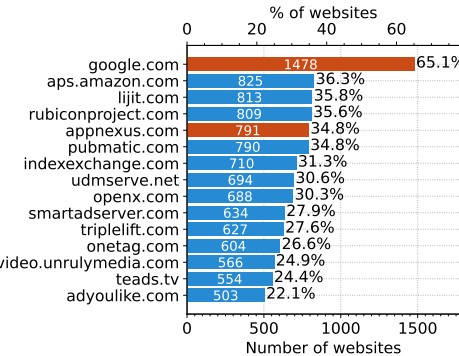

**Figure 10: Direct business relationships with the most unreliable websites in July 2024 based on both `ads.txt` and `sellers.json` files.**

## 5.1 Direct Business Relationships

We set out to explore what is the state of the advertising ecosystem two years after the Strengthened CoP was signed. We collect the `ads.txt` files of misinformation websites and the `sellers.json` files of ad networks on July 2024. We plot in Figure 8 the number of direct business relationships among misinformation websites and ad networks. This relationships are extracted based on the ad accounts that both publishers and ad networks agree are valid through `ads.txt` and `sellers.json` files respectively. We observe that the majority of misinformation websites have a direct contract with only a handful of ad networks. However, there exist misinformation websites that form business contracts with tens or even hundreds of ad networks. One such example is the website *whatreallyhappened.com*. This website serves an `ads.txt` file of over 8500 lines that contains verified `DIRECT` entries for 196 distinct ad networks. In fact, the identifiers in this file are organized and labeled accordingly, indicating the presence of (dark) pools [20, 24].

Next, we plot in Figure 9 the direct business relationships of top ad networks with misinformation websites. We observe that for most ad networks, the behavior is similar and have a DIRECT business relationship with about 16.9%-58.1% of misinformation websites whose `ads.txt` files we were able to obtain. Even though these values might seem small, the underlying implications are significant. They indicate that the most popular ad networks have a direct business contract with a large percentage of misinformation websites, and that they facilitate the proliferation of misinformation content through ad revenue. In fact, event the smallest value in Figure 9 suggests that ad networks have authorized at least 1 out of 5 misinformation websites that serve an `ads.txt` file. The two ad systems that stand out are Google, and Underdog Media that controls *udmserve.net*. These two ad networks have a direct business relationship with over 400 misinformation websites according to their `sellers.json` files. Interestingly, both ad networks disclose in their guidelines that they do not allow the monetization of misinformation content or content that makes false claims [31, 32], but evidently they are not able to filter out all of these websites. Altogether, there are hundreds of misinformation websites that still have direct business relationships with legitimate and popular advertising networks and are able to generate revenue from

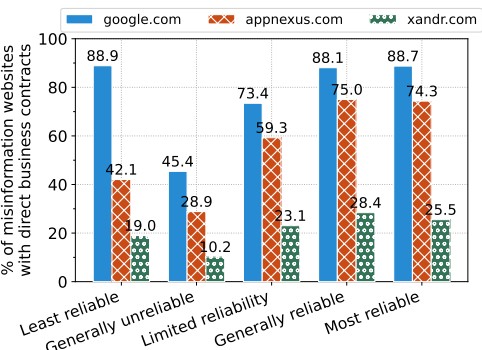

**Figure 11: Business relationships among signatories of the CoP and news websites of various credibility ratings.**

dis-/mis-infomration content. The accounts used to sell the ad inventory of these websites are recognized by popular ad networks.

> **Finding 2:** To this day, the most popular ad networks still have documented business relationships with over 500 distinct misinformation websites. Ad networks that have signed the CoP have approved as ad inventory sellers more than 1 out of 3 misinformation websites that serve an `ads.txt` file.

## 5.2 News Websites Credibility

To further support our finding that ad networks still work with misinformation websites to a large extent, we utilize the Science Feedback dataset on news websites credibility [33]. The researchers behind this dataset have accumulated a list of over 20K news websites from more than 100 countries. For each news website they assign a score reflecting its credibility, aggregating pre-existing and publicly available evaluations from trustworthy entities (i.e., academic researchers, fact-checking organizations, expert raters).

We access the original list and successfully fetch the `ads.txt` files of ∼7K news websites in July 2024. We perform a similar analysis as described earlier, but here, we make use of the list's original classification system and focus on websites labeled as "Generally unreliable" and "Least reliable". We are able to extract 2,459 unreliable news websites with at least one `DIRECT` record in their `ads.txt` files and present our findings in Figure 10. We discover a similar behavior with our previous findings. Popular and legitimate ad networks have a direct business contract with hundreds of unreliable news websites and this relationship is recognized by both the websites and ad networks. Finally, in Figure 11 we focus on the signatories of the Code of Practice on Disinformation and explore their direct business relationships with news websites of various credibility scores. We find that these ad networks work with all news websites and do not seem to have a distinctively different policy for unreliable news websites. In the case of Google, we find that it works with almost as many of the least reliable websites (percentage wise) as with the most reliable.

> **Finding 3:** Two years after the Strengthened Code of Practice on Disinformation was signed, even the ad networks that signed the CoP do not considerably distinguish between unreliable and reliable news sources, allowing publishers to generate ad revenue from false claims.

| Entity | % misinformation websites | # misinformation websites |
|---|---|---|
| Google LLC | 69.39% | 272 |
| RevContent, LLC | 20.66% | 81 |
| AdRoll, Inc. | 12.50% | 49 |
| RTB House S.A. | 12.50% | 49 |
| Microsoft Corporation | 8.93% | 35 |
| Criteo SA | 7.91% | 31 |
| MGID Inc | 7.14% | 28 |
| Taboola, Inc. | 6.12% | 24 |
| Zeta Global | 5.87% | 23 |
| Outbrain | 5.61% | 22 |

**Table 2: Ad network operators serving programmatic ads to misinformation websites. Even though Google and Microsoft have signed the CoP, they still deliver ads, and therefore revenue, to misinformation websites.**

## 6 SERVED ADVERTISEMENTS

In previous sections, we demonstrate that the most popular and legitimate ad networks still have business relationships with misinformation websites, and that the CoP did not have a substantial impact on the monetization of misinformation. To validate our observations, we study the ads that are actually served on misinformation websites. Our goal is to explore to what extent ad networks still monetize misinformation content and if reputable advertisers might end up next to misinformation content. Previous work has demonstrated the misinformation websites are short-lived [34], and therefore, the analysis of the following sections is based on active websites only. We analyze the ads found in active misinformation websites, based on the methodology described in Section 3.3. We describe the process of discovering active websites in Appendix C.

### 6.1 Ad Networks

Using data coming from 6 distinct user profiles (2 genders and 3 countries) and one clean user profile, we detect ads from 2,480 advertisers in 392 misinformation websites. We attribute the large number of distinct advertisers to the use of different vantage points (i.e., 3 geographic locations), thus being served ads from local brands.

First, we study the ad networks that deliver ads to misinformation websites. We achieve this by studying the ad URLs the crawler detected in any of the used personas. We analyze these ad URLs and extract the domain from them, grouping them on the eTLD+1 level. For example, ads served by *adclick.g.doubleclick.net* and *googleads.g.doubleclick.net* are all grouped under the *doubleclick.net* domain. Finally, to understand the actual entities involved in serving these ads, we group domains based on the companies that own these domains. For example, ads served by *doubleclick.net* and *googleadservices.com* are all attributed to Google LLC. To identify the entities that own and operate each domain, we utilize the Duck-DuckGo Tracker Radar dataset [35]. We acknowledge that there might be various intermediaries involved in the (re)selling of ad inventory but this analysis is outside the scope of this work. We summarize our findings in Table 2.

We find that there are a lot of legitimate and popular ad networks that still deliver ads to misinformation websites. We find that Google is the most prominent ad facilitator of misinformation websites serving ads to 69% of the misinformation websites, which contained ads. The second most popular ad network that

serves ads to misinformation websites is RevContent. Previous work has already demonstrated that RevContent is popular among misinformation websites [19, 36]. Unfortunately, we find that ads are served on misinformation websites even by entities that have signed the Code of Practice on Disinformation (e.g., Google and Microsoft) [29], suggesting that there is still work to be done.

It is important to mention that these findings highlight even more the issue described in previous sections with the analysis of `ads.txt` and `sellers.json` files. An ad network could argue that they had indeed authorized misinformation websites to enter their platform in the past (i.e., findings of `ads.txt` and `sellers.json` analysis) but they have now blocked them and no longer work with them. However, the findings of this analysis are definitive proof that some ad networks still have business relationships with misinformation websites. These ad networks serve the actual ads and eventually pay money to publishers of misinformation content.

To make matters worse, we discover that there exist ad networks that deliver ads of misinformation websites to users. For example, when our female USA-based persona visited the website *theconservativebried.com*, Google served an ad for *goop.com*, a website associated with pseudoscience and dozens of failed fact-checks [37]. We observe similar behavior for various ad networks, including Google, Outbrain, RevContent and Yandex LLC. Evidently, ad networks do not properly review the content they promote, thus disseminating misinformation. It is important to notice that the CoP explicitly states that signatories should work towards stopping ads that contain false claims (Commitment 2: "TACKLING ADVERTISING CONTAINING DISINFORMATION") and Google is one of the signatories. All the above suggest that there even though the CoP is a step towards the right direction, more effort is required to tackle the monetization and propagation of online misinformation.

> **Finding 4:** We discover that the most popular ad networks do not closely examine their ad placements. They still still serve ads, and thus ad revenue, to almost 400 misinformation websites. Google and Microsoft, signatories of the CoP, place ads in 70% and 9% (resp.) of misinformation websites in our dataset.

### 6.2 Brand Safety of Advertisers

Next, we analyze the actual advertisers that appear next to misinformation content. We utilize the dataset of the previous section but now focus on the landing pages users arrive when they click on ads next to misinformation content. We find that there exist a lot of advertisers whose ads are placed in dozens of different misinformation websites. This behavior is consistent for various types of ads and for all personas. For example, using the USA-based female persona, we find that *jjshouse.com*, a popular online store for wedding dresses was advertised in 80 distinct misinformation websites. Similarly, the Greece-based male persona was served ads from Temu in 86 distinct misinformation websites (22%) while ads from *overnightglasses.com* were served to the male US-based user in 59 distinct misinformation websites. We should note that this finding is not a finger pointing exercise for these advertisers and does not indicate that these advertisers choose to appear next to misinformation content. Most likely, these ads are placed next to misinformation because of the ad network believing they are of interest to the virtual user. However, we should note that this can have

a big impact on advertisers (i.e., brand safety) if visitors associate misinformation content with specific services or products.

Finally, we analyze the significance of the advertisers found in misinformation websites. To our surprise, we find ads from very reputable companies appearing next to misinformation content. We provide screenshots of some examples in an anonymous repository [38]. We find 23 Fortune 500 companies advertising in misinformation websites, threatening their reputation and consumer trust due to negative brand associations.

> **Finding 5:** Ads from very popular and renowned brands (even from 23 of the largest companies in the world) are served next to misinformation content. Users can see an ad of a specific brand in up to 86 distinct misinformation websites.

## 7 RELATED WORK

In [39], authors studied fake news and low-quality news websites and found that fake news publishers monetize their content using reliable ad servers. In [36], authors discovered that misinformation websites rely on popular Web services such as Cloudflare, Google and Revcontent to host their content and monetize it via ad revenue. In [40], authors studied the monetization schemes of misinformation websites, focusing on anti-vaccine pages, demonstrating the mainly rely on ads. In [41], authors performed a cost-based analysis on fake news interventions and proposed a model that can help combat the proliferation of misinformation websites. In [42], authors proposed a content-agnostic fake news detection service using network and structural characteristics, including requests towards third-party advertising services. In [43], authors utilized non-perceptual features (e.g., domain name, DNS config) to train a multi-class model that detects disinformation websites in the wild. In [34], authors found that some popular third-party services have higher presence in fake news sites compared to real news. In [44], authors compared advertising prices for desktop and mobile users and showed how these prices have grown over the years.

In [23], authors performed a longitudinal study of `ads.txt` files and their adoption. Similar to our work, in [19], authors fetched and processed `ads.txt` and `sellers.json` files to study the business relationships between fake news websites and ad networks. They discovered that fake news websites rely on popular and credible ad networks to generate revenue. In [20], authors measured the compliance of the ad ecosystem with ad standards. They explored how publishers use `ads.txt` files to form dark pools, a way to circumvent restrictions and monetize misinformation content. Similarly, in [24], authors further explored dark pooling along with hidden intermediaries and highlighted the importance of limiting the ad revenue generated by misinformation publishers. In [30], authors demonstrated that dark pooling can be diminished by notifying stakeholders and ad networks.

In [45], authors studied the content of ads served on either real or fake news websites and found that almost half of the ads they labeled were problematic. In [46], authors demonstrated that ads largely finance misinformation websites and that can have a great effect on advertisers' brand safety. In [47], authors performed a user study to understand how questionable content affects the brand of pre-roll advertisers. Similarly, in [48], the authors studied the effect of website quality on advertisers that utilize programmatic advertising. In [49], authors interviewed actual stakeholders involved in online advertising and explain that some advertisers might tolerate fake news websites because they can reach users at a lower cost. In [50], authors used a misinformation website with ads in Facebook to study how different generations interact with misinformation. Finally, previous work has studied disinformation campaigns that utilized Facebook ads to affect elections [51, 52].

## 8 DISCUSSION

The Strengthened Code of Practice on Disinformation is a step towards the right direction, but there is still work to be done. The goal of this work is not to discredit the effort of the CoP but rather to highlight the importance of scrutinizing ad placements and marking more misinformation websites as ineligible for monetization. We believe that ad networks should work towards ensuring that misinformation is no longer profitable and that even popular misinformation websites can no longer survive because of ad revenue. Reducing the financial incentives of misinformation spreaders can increase the quality of information and trust in factual reporting. Finally, the self-regulatory nature of the CoP is a positive indication of desire for a more reliable news ecosystem but unfortunately its adoption is limited, with only 44 signatories of various categories (e.g., ad networks, fact checkers, social media platforms) as of July 2024. It is evident that current signatories have been laying the foundation, but we hope new signatories from the online ad ecosystem (i.e., ad networks) will take initiative for more substantial actions.

It is customary in such studies to have a control group - that is, a set of non-disinformation websites and study what is the change in advertisements in these websites as well. Such a control group enables researchers to argue whether a change they see in misinformation websites is due to the CoP. However, this is not needed for this work. Indeed, we are not interested in attributing any changes we see to the CoP (or not). The goal of this work is to discover if the change that the CoP expected (i.e., that disinformation websites will not have ads) happened and to what extent.

## 9 CONCLUSION

The Strengthened Code of Practice on Disinformation was signed in June 2022, in an attempt to fight disinformation in a self-regulatory manner. The first commitment of the CoP is that entities involved in ad placements demonetize dissemination of disinformation. The expectation is that since the CoP was signed by leading industry players (e.g., Google and Microsoft), two years after the introduction of the Strengthened CoP, things will have improved. In this work, we perform a historical analysis and demonstrate that this is not the case. In fact, misinformation websites are still able to retain business contracts with popular ad networks and monetize their content. We show that the adoption of the CoP has not made a significant impact on the misinformation ecosystem, and that ad networks stopped working with only insignificant misinformation websites that do not have a lot of visitors. Finally, we perform a dynamic analysis of ads served on misinformation websites. We establish that not only popular ad networks still facilitate the monetization of the majority of misinformation websites, but that they also serve ads of respectable and popular brands next to misinformation content.

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

## A  USER PROFILES

To form a list of websites that are associated with a gender of a specific country, we utilize the SimilarWeb analytics platform [28]. Specifically, we visit SimilarWeb and extract the 5 most popular websites in each category and each country of our experiment. SimilarWeb uses a taxonomy system with 210 distinct website categories (including subcategories). For each website, we extract its demographic data (if available). SimilarWeb uses two gender groups (i.e., female and male) to classify visitors of websites. Next, we sort websites based on their audience's gender representation and ensure that all websites have a representation of at least 55% for that gender. We set an upper threshold of 200 websites and a lower threshold of 100 websites to train each persona while all. The lower threshold is more than enough since previous work has demonstrated that a persona is stable after visiting the top 10 websites associated with users of that demographic [53]. This process leads to 6 (combination of 2 genders and 3 countries) lists of website that represent the interests of the specific persona.

We utilize Playwright [54], a browser automation framework and set up legitimate browser profiles, retaining cookies and local storage. To build the user profiles, we try to emulate the behavior of a real user that visits some websites to consume online content. We try to make the behavior of our automated crawler as non-deterministic as possible to appear as a legitimate user. The behavior we are emulating is that of a user that visits a website, reads its content and then navigates through the website after quickly inspecting the content of each page. For each website in our target list we (i) visit the landing page, (ii) wait for a random number of seconds in range [2, 5], and (iii) explore 5 subpages. In order to explore a subpage our crawler clicks on a randomly selected internal URL and navigates to this subpage, waits for a random number of seconds to emulate a user consuming content, scrolls for a random number of seconds and then waits for an additional random number of seconds. The websites of the list are visited in a random order and after all websites have been visited, the user profile is stored.

## B  POPULAR MISINFORMATION WEBSITES

To our surprise, we find that even today, there are a lot of ad networks working with certain misinformation websites that one can hardly argue they are legitimate. For example, we find that there are at least 10 of the most popular ad networks (including Google, Amazon, AppNexus and Pubmatic) that have an explicit and direct business relationship with *worldlifestyle.com*. According to Media-Bias/FaceCheck, WordLifestyle is "a strong pseudoscience website based on the promotion of false or misleading claims regarding science and health" and it has a low factual reporting score "due to a lack of sourcing and blatant clickbait abuse to generate advertising revenue" [55]. Even such websites that are known to take advantage of the ad ecosystem in order to monetize misinformation still have direct business contracts with popular ad networks. In fact, by manually visiting this website, we find that Google indeed serves ads to visitors.

In addition to this, we find that the website of "One America News Network" (OANN) is an approved ad inventory seller within the most popular ad networks (including Google and Appnexus, signatories of the Strengthened Code of Practice on Disinformation). OANN is considered a far-right biased news website that promotes propaganda, conspiracy theories and has over 10 failed fact checks [56]. According to SimilarWeb, this website achieves over 2.5M monthly visits, and in conjunction with the fact that almost 95% of the visitors come from the United States, an advertising valuable country, it can generate substantial ad revenue.

# C ACTIVE MISINFORMATION WEBSITES

Inspired by previous findings that misinformation websites might have a short longevity [34], we aim to refine our list by accumulating only misinformation websites which are still active. Towards this extent, we perform a manual analysis using independent reviewers. We start from the extended list of 2,469 misinformation websites and using an automated crawler we take screenshots of the entire landing page of each website. Our crawler was able to successfully access 2,049 websites and capture screenshots. Next, two independent reviewers coming from a computer science background (not the authors of this work) evaluate which websites are still active based on the content of the landing pages. Reviewers are allowed classify each website as either "Active" or "Inactive". Moreover, reviewers are allowed to manually visit websites and navigate through its pages if they are not certain which label to assign simply based on the screenshot of the landing page. The two reviewers are independent of each other and completed the task in a different environment, without influencing one another.

While manually rating websites, the reviewers noticed that some domains were active in the sense of responding to requests, but no longer served news content. This was mainly because these websites were up for sale (i.e., parked domains) or because they served empty content. In both of these cases, reviewers classified such websites as inactive. Domains that did not respond to HTTP(S) requests, domains that were non-existent (i.e., DNS NXDOMAIN) and domains that timed-out when accessed were all labeled as inactive.

Both reviewers evaluated evaluated the entire list of 2,469 misinformation websites and agreed to include 1,889 websites, agreed to exclude 568 websites and disagreed on 11 websites. The reviewers agreed in 99.55% of websites and had a Cohen Kappa score of 0.99. Note that a high inter-rater agreement score is expected in this context because deciding if a website is active is an easy task for a human evaluator. We decide to only accept websites that both raters agree are active and end up with a list of 1,889 active misinformation websites. We make this list of active misinformation websites publicly available [14].

# D ETHICAL CONSIDERATIONS

In this work, we make concentrated effort to not affect the performance of any Web service in any way. To download ads.txt and sellers.json files, we utilize existing crawlers and reach each domain only once to download the specific file. This behavior is far less intrusive compared to the automated processes that DSPs and SSPs use daily to verify programmatic advertising. Additionally, our data was collected once and all of our analysis is performed offline. Regarding the collection of ads, we ensure that we click on each ad URL only once, similarly to methodologies presented in previous work [19, 30]. Since previous work has demonstrated that the cost per thousand ad impressions is only a few dollars [44], we argue that our study's influence on the advertising ecosystem is negligible. Finally, following the GDPR and ePrivacy regulations, we do not collect or process any personal information of real users.

