# OpenReview forum: "Before & After: The Effect of EU's 2022 Code of Practice on Disinformation"
_ACM.org/TheWebConf/2025/Conference — WWW 2025 Oral_

### Official Review · Reviewer_uUMb · 2024-11-11

**Novelty:** 6
**Technical Quality:** 5

**Review:**

This paper provides the first investigation of the impact of the Strengthened Code of Practice on Disinformation on the demonetization of misinformation websites. It shows an interesting finding that the adoption of CoP has a very limited impact on the misinformation ecosystem meanwhile it has a significant impact on the misinformation websites that are not popular. Popular ad networks still facilitate the monetization of misinformation, highlighting the importance of scrutinizing ad placements and identifying misinformation websites.

Pros:

- This work studies a novel and important problem of the impact of the Strengthened CoP on misinformation demonetization.
- The study design and rationales generally make sense. It provides a novel pipeline for studying similar problems.
- Findings are interesting and sometimes, surprising, and could be useful for policymakers.

Cons:

- The major concern is the whole study design and therefore findings are based on observational data, leading to correlation, not causation. Therefore, it is possible that the findings are opposite to what was found by the authors due to hidden confounding bias. There are extensive works using human behavior data collected online to study causal effects, like [1,2]. Causality can provide more rigorous analyses and exact causal effects.
- Disinformation and Misinformation are different, but the paper does not differentiate these two.
- Writing should be improved: Authors are suggested to add more illustrative figures to help explain the methods. For Related Work, instead of listing each work using the same format, it would be more helpful for readers if different works were compared, to explain how newer works were built from previous works.
- While the problem is identified, the potential solutions are still lacking. Authors may consider providing some insights on how to address the identified issues.

[1] Saha, K., Sugar, B., Torous, J., Abrahao, B., Kıcıman, E., & De Choudhury, M. (2019, July). A social media study on the effects of psychiatric medication use. In Proceedings of the International AAAI Conference on Web and Social Media (Vol. 13, pp. 440-451).
[2] Cheng, L., Guo, R., & Liu, H. (2022, February). Estimating causal effects of multi-aspect online reviews with multi-modal proxies. In Proceedings of the Fifteenth ACM International Conference on Web Search and Data Mining (pp. 103-112).

**Questions:**

What are some potential ways to improve this proposed issue?

**Reviewer Confidence:**

3: The reviewer is confident but not certain that the evaluation is correct

**Scope:**

4: The work is relevant to the Web and to the track, and is of broad interest to the community

---

### Official Review · Reviewer_yzmi · 2024-11-13

**Novelty:** 6
**Technical Quality:** 6

**Review:**

The manuscript investigates whether EU's 2022 Strengthened Code of Practice had an effect in demonetizing well-known disinformation websites by stopping ad networks from supplying ads to those sites.

Strengths

S1. This is a problem/question of extreme relevance (are the current self-regulatory mechanisms working?) that hasn't been investigated (to the best of my knowledge).

S2.  The paper provides 5 interesting findings, which are not common knowledge.

S3. The authors use a variety of methods to accomplish the task, all of which are technically solid.

S4.  The analysis provided in the paper is comprehensive and the conclusions are adequately substantiated by evidence.

S5.  The paper is well-organized and well-written (although the Related Work section can be improved).

S6. They curated a list of misinformation news websites from various sources and made it publicly available.


Weaknesses

W1. The paper does not list its limitations explicitly. In particular, the question of financial incentives may never be properly answered without data on the disinformation websites' ads revenue. For example: It might be the case that ad networks are paying less for ads on these websites that they used to (before the Code of Practice). If this is true, then these sites are being demonetized even though they are still in business with the same ad networks.

W2. The absence of data from 2018 raises questions as to whether there was an overall decrease in the amount of advertising relationships (including for the popular websites) after the first Code of Practice was released (in 2020) that is just not observed in the study.

W3. The Related Work section reads as a sequence of statements of the type "[XX] did this; [YY] did that" without a clear and cohesive grouping of the references. Moreover, the section doesn't highlight the differences between the manuscript and the previous work. Particularly, as stated by the authors, [9] performed a similar analysis on fetch ads.txt and seller.json. It would be helpful to clarify how your work differs from theirs.

W4. Section 5.2 uses data on news websites credibility to further support the finding that ad networks still work with misinformation websites. It is clear that categories. "Generally unreliable" and "least reliable" are related to whether a website conveys false information. However, the authors do not comment on the overlap between these lists.

W5. The idea of using distinct personas is good, but 1/3 are US-based and 2/3 are Europe-based. The authors do not discuss whether it was useful to include 2/3 from Europe and whether not including Asia, Latin America or Africa would have an impact on the conclusions.

W6. Minor presentation issues (see comments below).

**Questions:**

Questions

Q1. Can you discuss the limitations of the work (in particular W1)?

Q2. Is it possible that there was a significant decrease in the advertising relationships between 2018 and 2022 that is just not observed in the data? If not, is there evidence that shows this?

Q3. Did you find some overlap between the two datasets? If you remove the overlap, are the conclusions still valid?

Q4. In section 4 Regulation Impact, shouldn't D be defined as $D = \frac{B-A}{|B|}$ to show the **decrease**?


Minor/typos

- Table 1, column labels are unclear, especially 'minimal'. Also, for p-value, 3-4 decimals is enough.
- Table 2 contains data from 2024. It would be helpful to specify this timeframe in the caption.
- ‘This relationships’ -> These relationships’
- ‘In fact, event the smallest’ ->‘In fact, even the smallest’
- ‘dis-/mis-infomration content’ ->‘dis-/mis-information content’
- ‘All the above suggest that there even though the CoP …’ -> All the above suggest that even though the CoP…’

**Reviewer Confidence:**

3: The reviewer is confident but not certain that the evaluation is correct

**Scope:**

4: The work is relevant to the Web and to the track, and is of broad interest to the community

---

### Official Review · Reviewer_vRTb · 2024-11-22

**Novelty:** 5
**Technical Quality:** 5

**Review:**

This paper looks into the impact of the Strengthened Code of Practice on Disinformation, addressing an important and timely issue. By examining how well the existing framework works, it provides helpful insights for policymakers to improve future policies. The authors take an innovative approach, showing the importance of reviewing current measures to make them more effective.

Strengths:
- The paper is well written and easy to follow.
- Using different personas to evaluate the results was a thoughtful and effective approach.

Weaknesses
- The related work section is poorly written. There is no narrative, or a research gap highlighted based on the past literature.
- The study lacks discussion on how these findings can guide policymakers in creating more effective policies in the future.

**Questions:**

- Authors have used disinformation and misinformation interchangeably. Disinformation and misinformation are two different things. ''Misinformation refers to the inadvertent spread of falsehoods, regardless of intent, while disinformation refers to spreading falsehoods with the deliberate intent to mislead'' [1]. I suggest that they clearly define the terms and use the correct term for better clarity.
- The authors mention a misinformation website dataset they used. It would be beneficial to understand how representative this dataset is to have a better understanding of the results.
- Finding 6 reports that ads of 23 Fortune 500 companies are served next to misinformation content. I couldn't find an explanation of how the authors clarified whether certain content is misinformation or not. Is it misinformation content as in these websites had specific content which were misinformation and the adverts were positioned right next to those or they were positioned in websites which were classified as misinformation websites? If it was the former, I suggest that authors discuss how these contents were classified.
- Was there a specific reason why these specific geographic locations were used?
- There is a typo on Finding 4 (page 7): The word ‘still’ has been written twice.
- The authors discuss a potential effect of adverts being next to misinformation content would be the association of brand with these misinformation content. Is there any past literature that can support this claim?
- The related work section is poorly written. There is no narrative, or a research gap highlighted based on the past literature. I recommend that the authors revise the literature review to create a clear narrative that effectively leads up to the research gap they aim to address.


References:

[1] Scott Appling, Amy Bruckman, and Munmun De Choudhury. 2022. Reactions to Fact Checking. Proceedings of the ACM on Human-Computer Interaction 6, CSCW2 (2022), 1–17

**Reviewer Confidence:**

2: The reviewer is willing to defend the evaluation, but it is likely that the reviewer did not understand parts of the paper

**Scope:**

3: The work is somewhat relevant to the Web and to the track, and is of narrow interest to a sub-community

---

### Official Review · Reviewer_riGX · 2024-12-02

**Novelty:** 4
**Technical Quality:** 4

**Review:**

Thanks for submitting your work to WWW. I appreciate the idea of this research; it's timely and meaningful, especially when it attempts to assess the impact of CoP on disinformation. Regulations like CoPs and their efficacy evaluations are necessary to combat growing misinformation and disinformation campaigns. However, some concerns need to be addressed to strengthen the paper further.

1. Lack of Clarity: There is no clear structure or a high-level description of the overall approach, followed by the technical details. Instead, it seems like a section is coming after a section without any concise explanation of why the reader is seeing that section or how the analysis ties to the overall story of the paper.

2. Ground truth verification missing: The paper uses the MediaBias/FactCheck (MBFC) website and extracts data from it to identify potentially deceiving websites. Since the data source is not peer-reviewed and provides no details on how it classifies websites into different categories, authors must ideally select a random subset of the websites and manually verify their accuracy.

3. Novelty (a clear departure from the related work missing): The paper cites several interesting papers, including [39], that study fake news and low-quality news websites. However, the paper does not mention why it does not use the websites mentioned in [39]. Moreover, the [39] paper mentions that `` Our study reveals that fake and low-quality publishers demonstrate a higher tendency to serve more ads and to partner with risky ad servers than traditional news media
with similar popularity and age.’’ This seems quite relevant to this paper. Also, it uses Zimdars list to curate fake websites. Why didn’t this paper also use the same list? Thus, the authors must clearly show the research gap. The same applies to other cited papers as well.

Please find more additional comments here:
1. Please provide high-level details about the methodology in the introduction; it provides more context to the paper (a suggestion!)


2. “We also filter out Facebook because we argue that it is neither a news publisher or
news aggregator.” ---> Leaving out big players like Facebook and Twitter should be reconsidered, as this is where the major issues could be. Such organizations should not get an easy pass because the analysis is complex. The authors could consider specific posts and relate them to the ad ecosystem.

3. In the background, provide a realistic example mentioning website A ad.txt mentions Y as the advertiser and Y’s seller.json mentions A as the seller. It's confusing without the exampleof  the precise role of ad.txt and seller.json.

3. “We crawl the list of approximately 2,500 misinformation websites on January 2024 and download 226,659 ads.txt entries coming from 1,132 misinformation websites.’’---> Its difficult to understand how there are almost 100x ads.txt than websites. Does a website contain more than one ads.txt?

4. How personas are made is fundamental to your evaluation. Please provide some critical details on how the authors created the personas in the main paper itself. The details are mentioned in Appendix, but not clear. For instance, “Next, we sort websites based on their audience’s gender representation and ensure that all websites have a representation of at least 55% for that gender.’’ Its very difficult to parse such statements.

5. Authors use consent-o-matic to automatically accept cookie banners on websites. However, a recent work [Ali et al.] claims the BannerClick (https://bannerclick.github.io/) tool is 99% efficient in accepting the banners; the authors also show that their banner coverage is higher than that of the content-o-matic tool. Please refer to this publicly available tool to accept the banners for your analysis.

6. “First, we study the behavior of the top ad networks and how their business relationships with misinformation websites have evolved over a period of 2 years.”---> Please provide steps on how you measured the evolution.

**Questions:**

1.) Can you elaborate on the accuracy of the ground truth, i.e., MBFC website? Also, why didn’t you use other sources e.g., Zimdars list?

2.) Please explain how your work is different compared to the existing work.

3.) Please clarify on numerous gaps with missing details as pointed out in the review.

4.) Can you provide a high-level overview of your approach and improve your writing?

**Reviewer Confidence:**

3: The reviewer is confident but not certain that the evaluation is correct

**Scope:**

4: The work is relevant to the Web and to the track, and is of broad interest to the community

---

### Official Review · Reviewer_sH7j · 2024-12-02

**Novelty:** 6
**Technical Quality:** 6

**Review:**

This work examines the impact of the Code of Practice on Disinformation (CoP) on one of its proposed commitments: the demonetization of disinformation websites. Specifically, the study analyzes how extensively misinformation websites continue to be funded through advertisement revenue two years after the CoP's implementation. To achieve this, the authors compare the funding of misinformation websites during a period prior to the CoP’s introduction (2021) and afterward (2024).
Using an interesting, bot-based approach, the main claim is that the CoP has, so far, had very little to no impact on the demonetization of misinformation websites. Despite the study's relevance, there are two key aspects of the work methodology that limit the authors' ability to infer some of the causal relationships they identify.

 1. Popularity of Websites: the authors conclude that the significant decrease in the number of direct business relationships between ad networks and misinformation websites observed in Figure 2 is primarily due to the dropout of ad networks from less popular misinformation websites. However, as far as I can  tell, the classification of a website's popularity is based on its traffic data from 2024 rather than from 2021. If the demonetization of websites influences their popularity, the observed lack of popularity of these websites in 2024 could be a consequence, rather than a cause, of their advertisement demonetization. In other words, having no audience can be both a cause and a consequence for the ad networks to stop their relationships with the websites.
In this line of reasoning, given that 461 misinformation websites included in the 2021 dataset still have direct relationships with ad networks in 2024 (Figure 2), it would be valuable to determine how many of the websites from the 2021 dataset—those without ads.txt files or no direct relationships with ad networks in 2024—remain active today and compare their past and current popularity. This analysis would also contribute to help measure the significance of advertisement support for this type of website.

2. Misinformation Website' Definition: The methodology for generating the database using Media Bias/Fact Check does not exclusively select misinformation/fake news websites. Under the selected "Low-credibility" classification from Media Bias, even that coupled with the factual reporting, will select/include websites such as FoxNews and other extremely biased but popular news outlets.
This distinction is crucial for interpreting the results and understanding whether the observed trends are truly representative of the real levels of misinformation website demonetization.
In summary, the pros and cons of the work are:

PROS:
* Relevance of the topic: it is  important to address whether measures like the CoP have a real impact in the decrease of misinformation websites or whether they need improvement to become more consequential.
* Conservative Analysis: the methodology is rigorous in examining relationships between websites and ad networks, as it selects only cross-referenced entity IDs in ads.txt and sellers.json. Furthermore, the additional step of collecting advertisements actually displayed on misinformation websites adds value by confirming that the identified relationships are real and not outdated.
* Ethical concerned data collection: by leveraging files such as ads.txt and sellers.json to investigate relationships between ad supply networks and website publishers, the authors minimized the number of visits to misinformation websites and displayed advertisements.
* Public available datasets: the authors share their list of misinformation websites, making the dataset accessible for further research.

CONS:
* Definition of misinformation websites: as mentioned above, the definition considered in the study may be too broad. It would be important to identify which are the popular websites that remain funded by the ad networks.
* Causal Inference: The claim that “popular ad networks still facilitate the monetization of the majority of misinformation websites” implies that a) the popular websites in 2024 are the same as in 2021; and b) that their popularity is independent of the received ads.

**Questions:**

* The key question is whether a causal relationship can be inferred, namely whether advertisers choose to only "drop" the low visibility websites and to continue placing ads on the most visited ones. The ideal experiment would be to compare the traffic of the same websites in 2021 and in 2024. Most of the following questions try to address this point.

Q1 - How much of the total volume of identified business relationships is covered by the top ad networks shown in Figures 2, 4, and 5? What was the criteria?

Q2 - From my understanding, the decrease in bar size in Figure 2 does not imply that a website has stopped having ad relationships entirely, but rather that there has been a shift in the ad network placing the ads. How many of the misinformation websites from 2021/2022 remain active today even if not funded?

Q3 -  Given the values in Figure 7, how prevalent is the dark pooling in the dataset? Can this contribute to ad networks not being able to identify with which website they are in business with?

Q4 -  Where the unpopular websites in 2024 already unpopular in 2021?

Q5 -  Why are the criteria used to select misinformation websites different than the ones used in the study providing the data for 2021/2022?

Q6 - What could explain the discrepancies among companies shown in Table 2 and Figure 9 (for instance)?

Q7 - Could the results in Figure 4 reflect the fact that ad networks do not classify such outlets as misinformation or disinformation websites?

Despite the paper's notable strengths, particularly in terms of the interesting approach and relevance of the question it poses, I believe the above concerns need to be addressed or the causal claims need to be substantial toned down.

**Reviewer Confidence:**

4: The reviewer is certain that the evaluation is correct and very familiar with the relevant literature

**Scope:**

4: The work is relevant to the Web and to the track, and is of broad interest to the community